# Anticonvulsant Activity of *Bombyx batryticatus* and Analysis of Bioactive Extracts Based on UHPLC-Q-TOF MS/MS and Molecular Networking

**DOI:** 10.3390/molecules27238315

**Published:** 2022-11-29

**Authors:** Qinglei Wang, Rong Wang, Cheng Zheng, Linlin Zhang, Hong Meng, Yi Zhang, Linke Ma, Bilian Chen, Juanjuan Wang

**Affiliations:** 1School of Pharmaceutical Sciences, Zhejiang Chinese Medical University, Hangzhou 310053, China; 2NMPA Key Laboratory of Quality Evaluation of Traditional Chinese Medicine (Traditional Chinese Patent Medicine), Zhejiang Institute for Food and Drug Control, Hangzhou 310052, China; 3Department of Pharmacological Toxicology, Zhejiang Institute for Food and Drug Control, Hangzhou 310052, China

**Keywords:** *Bombyx batryticatus*, anticonvulsant activity, different polar fractions, UHPLC-Q-TOF-MS/MS, molecular networking

## Abstract

*Bombyx batryticatus* (BB) is an anticonvulsant animal medicine in traditional Chinese medicine (TCM) and acts on the central nervous system. This research aimed to study the anticonvulsant effects of different polarity fractions of extracts from BB and to explore the components conferring anticonvulsant activity. Materials and methods: Crude extracts of BB at 20 g/kg were divided into different polarity fractions (petroleum ether, chloroform, ethyl acetate, water) and were administered to groups of mice before injecting pentylenetetrazol (PTZ) to induce convulsions. The animals were placed in chambers, and their behaviors were recorded for 30 min following the injection. Latency time, percent of protection, convulsion, convulsion rate, and convulsion score were determined for these mice. The compounds present in the different fractions were analyzed, and those from the fraction that conferred anticonvulsant activity were identified by high-performance liquid chromatography-quadrupole-time-of-flight mass spectrometry (UHPLC-Q-TOF MS) and molecular networking (MN). The chloroform extract fractions (B-C) clearly increased the seizure latency time and protection percentage and decreased the convulsion percentage compared to the control group. The anticonvulsant effect of other extract fractions was not significant. Our study shows that the chloroform extract fractions (B-C) of BB have a significant anticonvulsant effect. We also identified 17 compounds including lumichrome, pheophorbide A, and episyringaresinol 4′-*O*-beta-d-glucopyranose that were found for the first time. The results of this study may lay the groundwork for studying compounds derived from *Bombyx batryticatus* and their anticonvulsant effect.

## 1. Introduction 

Epilepsy is a neurological condition with a prevalence of 0.5–1% in humans and a lifetime incidence of up to 5% [1]. The incidence of epilepsy has shown an upward trend year by year in China, which is also a high-incidence area of epilepsy, where nearly half of the epileptic patients have refractory epilepsy [2]. Several causes are thought to induce epilepsy. For example, genetic predisposition, brain injury, stroke, or tumors [3], but the exact mechanism of epilepsy is still not well understood. Although there are a lot of anticonvulsant drugs available at present, many epilepsy patients are refractory to medications, and most anticonvulsant drugs cause some adverse effects [4]. The relevant literature reported that an increasing number of Traditional Chinese medicines (TCMs) have an anticonvulsant effect, which supports the usage of TCM in the treatment of epilepsy, further sustained by the lesser risk of side effects [5]. Therefore, it is a relatively new strategy to use TCM to treat epilepsy. Anticonvulsant effects were demonstrated for animal TCM, in particular for drugs obtained from *Bombyx batryticatus* [6].

According to its specific symptoms, epilepsy is considered to belong to the category of “convulsants” in traditional Chinese medicine. BB is the dried larva of *Bombyx mori* L. (silkworm of 4–5 instars) after infection by *Beauveria bassiana* (*Bals.*), recorded in the Chinese Pharmacopoeia. BB was reported to treat convulsions by Li Shizhen in the *Compendium of Materia Medica*. In traditional medicine, BB is considered to exert anticonvulsant, antiepileptic, neurotrophic, anticoagulant, antitumor, antibacterial, antioxidant, and hypoglycemic effects, among others. As such, BB has been used to treat convulsions, epilepsy, cough, asthma, headaches, skin prurigo, scrofula, tonsillitis, urticarial, parotitis, and purpura [7]. Previous studies revealed that BB has anticonvulsant effects in seizure tests in mice [6]. The anticonvulsant activity of BB has been verified in nikethamide-induced and isoniazid-induced seizure models, and the result was that beauvericin can significantly prolong the latent period [8]. Additionally, another study reported that ethanol extracts of BB had an obvious anticonvulsant effect on epileptic mice in which epilepsy was induced by maximal electroshock and Metrazol in dose-dependent and time-dependent manners [9].

Some researchers have studied the components of BB and tried to identify the bioactive ingredients. Protein, peptides, fatty acids, flavonoids, nucleosides, steroids, coumarin, polysaccharide [7], and other compounds have been reportedly isolated from BB [10], but their anticonvulsant activities have not been tested. Several papers suggest that beauvericin might be the active component of BB [11], while pharmacological experiments indicated that beauvericin has much less of an effect than a complex BB extract, which means there must be other undiscovered substances that play an anticonvulsant role together with beauvericin. Thus, the bioactive components of BB remain unclear, and it is essential to identify the compounds that have an anticonvulsant activity.

The traditional structural elucidation of new compounds is time-consuming and cumbersome [12]. Recently, molecular networking has become an increasingly popular tool. MN is a useful tool to analyze the data acquired by UHPLC-Q-TOF-MS, which is a computational strategy that may help the visualization and interpretation of complex data and find second-stage mass fragment ions and compounds that can be confirmed by fragmentation pathways [13]. MN can produce representative networks for molecular classes by using spectra from standards. Then, compounds with the typical fragments are combined in the representative networks, and by comparison with the standard networks, new compounds are found. MN is used to predict active compounds [14,15,16,17]. Therefore, the combination of UHPLC-Q-TOF-MS and MN is a powerful way to explore substances relevant to TCM [18]. The MN-guided approach could be used to develop novel compounds and analyze TCMs, as the composition of TCM is complex and diverse. The MN approach can help to intuitively find relationships among compounds.

In this work, we first studied the anticonvulsant effects of different polarity fractions of BB extracts. Then, we analyzed the active components in the fractions by UHPLC-Q-TOF MS and MN. The results from this study may lay the groundwork for future research on active compounds of BB and are seminal in exploring a new realm of anticonvulsant therapeutics.

## 2. Results and Discussion

### 2.1. Anticonvulsant Effects of Different Fractions on PTZ-Induced Seizures

The activity of the crude extract was first evaluated, and the result demonstrated that the crude extracts exhibited obvious anticonvulsive effects in a dose-dependent manner compared to the control and positive control medicine. The latency time of convulsion after administering the crude extracts at 10 g/kg, 20 g/kg, and 30 g/kg increased compared to those measured for the control group (Figure 1). All animals (*n* = 9) treated with 0.2% CMC-Na had convulsions. Five Animals (*n* = 9) treated with the crude extracts had convulsions. The results showed that the seizure latency time of the different dosages of extracts was not significantly different from that of the positive control group, whereas it was significantly different from that of the control group. The medium dose of the crude extracts showed a significantly different effect compared to the control (*p* < 0.05). The latency time of convulsion after administering the high dose of the crude extracts was also significantly different from that of the control group (*p* < 0.01).

After confirming the activity of the extracts, we examined the activity of the fractions. The latency time, protection, and convulsion rate measured for the different fractions in the PTZ experiment showed that the chloroform extract extracts (B-C) significantly increased the seizure latency time and the percentage of protection and decreased the percentage of convulsion compared to the control (Table 1). The seizure latency time of the group treated with B-C was 159.88 ± 19.67 s, whereas that of the control group was 60.09 ± 31.19 s, and that of the positive medicine group was 226.13 ± 34.87 s; the seizure latency time of the B-E group was 78.50 ± 21.87 s, that of the B-W group was 140.00 ± 39.34, and that of the B-P group was 95.20 ± 30.93 s. The percentage of protection against seizures induced by B-C was 46.15%, and that measured in the control was 0% (Table 1). The chloroform extract fractions showed the highest anticonvulsant activity among the different extract fractions (Figure 2). The percentage of protection measured for the chloroform extract fractions (B-C) showed that they provided significant protection against PTZ-induced mortality in mice (Figure 3).

Pentylenetetrazol-induced kindling is an experimental model for epilepsy. Pentylenetetrazol is a GABAA receptor antagonist commonly used as a convulsing drug in experimental studies [19]. The PTZ model is nonselective with respect to the mechanisms of seizure, so it is well suited for screening anticonvulsants [20].

We compared the anticonvulsant effects of the different fractions by examining the relationship between convulsion score and survival time. We achieved seizure scores of 1, 2, 3, and 4 at different latency times for the control, ethyl acetate fraction, petroleum ether fraction, water-soluble fraction, chloroform fraction, and a positive control fraction (Figure 4). Mice administered the chloroform fraction were the last to undergo convulsion, which means the chloroform fraction had an appreciably anticonvulsive effect.

### 2.2. Molecular Network Based on the Compounds from Different Polarity Fractions

To identify the active anticonvulsant compounds, the LC-MS data (Figure 5) from four different polarity fractions of the extracts were analyzed by molecular networking through GNPS (https://gnps.ucsd.edu/ProteoSAFe/status.jsp?task=05f5ed5f0a28414fb3f257ff37232be1 (accessed on 6 April 2022)). Compounds were mostly found in the chloroform extract fractions compared to other extract fractions, whose spectra are shown in different colors. The petroleum ether extract fraction is shown in pink, the chloroform extract fraction in blue, the ethyl acetate extract fraction in green, and the water extract fraction in orange. The molecular network contained 762 molecular features and 34 independent clusters with at least three features. A total of 56 compounds from four the different polarity fractions were matched through the MS/MS library of GNPS. Within the 56 compounds, 17 compounds from the chloroform extract fraction were analyzed (Table 2), and two compounds that were only detected in the chloroform extract fractions were analyzed by UHPLC-MS/MS and MN (Figure 6).

### 2.3. UHPLC Q-TOF-MS Profiling of Chloroform Extracts of BB

The analysis of the chloroform extracts of BB was performed by the previous 2.5 UHPLC-MS/MS method (Figure 7). The identification of the compounds was enabled by comparison of the MS spectra and MS/MS spectral data with those in MN, PubChem, other spectral libraries, and reference materials reported in the literature. Sixteen compounds were identified from the chloroform extracts of BB (Table 3). These compounds included peptides, lipids, and amides.

Peak 13 (RT: 33.9) was characterized as Bassianolide (11), with the precursor ion [M + NH_4_]^+^ *m*/*z* 26.6439 at 33.9 min, presented fragments with 210.1490 *m*/*z*, 909.6165 *m*/*z*, 228.1595 *m*/*z*, and presented fragments similar to the GNPS database. A precursor ion at the retention time of 31.082 min with [M + Na]^+^ *m*/*z* 806.4002 showed fragments at *m*/*z* 134.1311, *m*/*z* 244.1331, *m*/*z* 784.4168, and *m*/*z* 262.1437 and was characterized as beauvericin (9) by comparing to masses of the GNPS database. Beauvericin A (10) had a precursor ion with [M + NH_4_]^+^ *m*/*z* 815.4594 (RT: 32.158 min), which was confirmed by the fragments at m/z 244.1330, *m*/*z* 134.0967, *m*/*z* 537.2946, *m*/*z* 262.1431. Both adults and larvae readily accept *N. glauca* and *D. stramonium* as alternative hosts to *N. tabacum*, *P. peruviana* and *S. origanifolia* and are able to complete their development on these host plants. Peak 9, 10, 11 indicated secondary metabolites of *Beauveria bassiana* induced by parasitizing *Bombyx batryticatus*.

BB contained abundant lipids. Loliolide (1) (RT: 15.1 min), with [M + H]^+^ *m*/*z* 197.1173, was characterized by fragments at *m*/*z* 133.1014, *m*/*z* 179.1058, *m*/*z* 161.0946. The fragmentation pattern of compound **1** was likely to be as that shown in Figure 8. Peak 6 showed a precursor ion with [M + H]^+^ *m*/*z* 279.1593 and was characterized as dibutyl phthalate (6), with a retention time of 26.5 min, which was confirmed by the presence of an intense fragment at *m*/*z* 149.0238, *m*/*z* 150.0247, *m*/*z* 279.1593.

The amides were less abundant than previously reported in the literature. Many amides were present in the chloroform extracts of BB. MS/MS revealed a precursor ion with retention time of 33.8 min at [M + H]^+^ *m*/*z* 256.2636, corresponding to the C_16_H_33_NO molecular formula. Fragments at *m*/*z* 130.1232, *m*/*z* 144.1374, *m*/*z* 102.0909, and *m*/*z* 158.1541 made it possible to characterize peak 14 as palmitamide. A precursor ion with a retention time of 34.6 min at [2M + H]^+^ *m*/*z* 563.5511 presented fragments at *m*/*z* 282.2783, *m*/*z* 265.2540, *m*/*z* 247.2422, and *m*/*z* 283.2822 and was characterized as 9-octadecenamide, (Z) (12). Peak 8 (RT: 28.4 min) analyzed as phytosphingosine showed a precursor ion with [M + H]^+^ *m*/*z* 318.3004 and fragments at *m*/*z* 282.2779, *m*/*z* 270.2780 and *m*/*z* 264.2667. Aurantiamide acetate (5) with [M + H]^+^ *m*/*z* 445.2135 was characterized by fragments at *m*/*z* 194.1168, *m*/*z* 224.1070, *m*/*z* 105.0330, and *m*/*z* 252.1019. Peak 14 (RT: 34.4 min) identified as oleoyl ethanolamide presented a precursor ion with [M + H]^+^ *m*/*z* 326.3060 and fragments at *m*/*z* 308.2951, *m*/*z* 309.2820, *m*/*z* 121.1021 and *m*/*z* 135.1178.

Octadecanamide was proposed for peak 16 (RT: 36.4 min) due to the presence of the precursor ion [M + H]^+^ *m*/*z* 284.2954 and the fragments at *m*/*z* 284.2951, *m*/*z* 285.2980, *m*/*z* 286.3013.

The compounds *N*-(2-phenylethyl) hexadecanamide (8), 3-(4-hydroxyphenyl)propanoate (2) episyringaresinol 4′-*O*-beta-d-glucopyranose (3), pheophorbide A (15) in *Bombyx batryticatus* were analyzed. Fragment ion masses and precursor ion masses were matched with the GNPS database.

Previous studies performed the identification of compounds by LC-MS/MS. The compounds were analyzed by high-resolution mass spectrometry. We analyzed for the first time compounds 3 and 4from *Bombyx batryticatus* [12,41,42,43,44].

## 3. Materials and Methods

### 3.1. Materials and Reagents

Pentylenetetrazol (PTZ) was obtained from Aladdin (Aladdin, Shanghai, China). Sodium chloride (BDH) was purchased by Zhejiang Kancheer Pharmaceutical Co., Ltd. (Dongyang, China). Chromatographic-grade methanol was purchased from Merck (Rahway, NJ, USA). Petroleum ether, chloroform ethanol, ethyl acetate, and sodium carboxymethylcellulose were purchased from Sinopharm Chemical Reagent Co., Ltd. (Shanghai, China). The insect sample was collected from Ping Yi country in the Shan Dong province and was then identified and authenticated by Guo ZengXi, a Chinese Medicine Practitioner.

### 3.2. Extraction of Bombyx Batryticatus

*Bombyx batryticatus* powder (50 g) was heated under reflux two times for 1 h each in 300 mL of 75% ethanol each time and then concentrated under reduced pressure until alcohol-free; a crude extract (C-E) was prepared. The C-E was sequentially extracted with petroleum ether, chloroform, and ethyl acetate. Powders of the different extracted fractions (petroleum ether (B-P), chloroform (B-C), ethyl acetate (B-E), water (B-W)) were obtained through evaporation of the solvent. The powders were dissolved in 50 mL of 0.9% NaCl containing 0.2% of CMC-Na.

For the pharmacological experiments, the resultant solution was further 15-fold diluted with methanol, and filtered the mixtures through a 0.22 μm microporous membrane filter after centrifugation, and finally analyzed them by UHPLC-Q-TOF-MS/MS.

### 3.3. Animals

The male institute of cancer research (ICR) mice weighing between 22 and 30 g used in this study were purchased from Shanghai Jihui Laboratory Animal Breeding Co., Ltd. (Shanghai, China). The animals were maintained under standard temperate conditions (22 ± 5 °C) in the animal house of the Zhejiang Institute for Food and Drug Control. All experimental procedures were approved by the Zhejiang Institute for Food and Drug Control. Ethical consideration approval was granted for this study with ethical clearance number: 001.

### 3.4. Experiment

#### 3.4.1. Effects of the Low, Middle, and High Doses of Crude Extracts on PTZ-Induced Seizures

Forty-five male ICR mice were divided into 5 groups (*n* = 9 animals in each group). The mice in these five groups were administrated a carboxymethylcellulose sodium solution (0.2% CMC-Na + 0.9%NaCl), carbamazepine (80 mg/kg), low crude extract (10 g/kg in crude drug), middle crude extract (20 g/kg in crude drug), and high crude extract (30 g/kg in crude drug) by intraperitoneal injection (i.p.); 30 min later, the mice were injected with PTZ (100 mg/kg, i.p.). The crude extracts of BB and PTZ were dissolved in 0.9% NaCl containing 0.2% CMC-Na.

After PTZ administration, all animals in the different groups were placed in chambers (31.5 cm × 20 cm × 12 cm), and their behavior was recorded for 30 min. Latency time, percentages of protection and convulsion, convulsion rate, and convulsion score were recorded. According to the Racine scale, the seizure score was determined (Table 3), and an attained score of 4 was considered as convulsions [45].

#### 3.4.2. Effect of Different Fractions on PTZ-Induced Seizures

Fifty-four male ICR mice were divided into six groups (*n* = 9 animals in each group). Group 1, carboxymethylcellulose sodium solution (0.2% CMC-Na + 0.9% NaCl), group 2, carbamazepine (80 mg/kg), group 3, B-P extract, group 4, B-C extract, group 5, B-E extract, and group 6, B-W extract, received the mentioned treatments 30 min before PTZ (100 mg/kg, i.p.) was administered. The dosage of all the extracts was 20 g/kg, and the experiment was conducted as described previously in Section 3.4.1.

### 3.5. UHPLC-Q-TOF MS Analysis

We used the chromatographic column Eclipse Plus C18 column (2.1 × 50 mm, 5 μm). The mobile phase was a binary solvent system consisting of solvent A (5 mM ammonium formate in water) and solvent B (methanol). Gradient elution was conducted as follows: 0–10 min (0–30% B), 10–20 min (30–73% B), 20–35 min (73–95% B). The column temperature was 30 °C, and the flow rate was 1.0 mL/min. The injection volume was 1 μL.

For the UHPLC-Q-TOF MS analysis of the compounds in the active fractions of the anticonvulsant, we used a UHPL chromatographer (UHPLC-1290 infinity Ⅱ Agilent) interfaced with a mass spectrometer (6545 Q-TOF, Agilent Technologies, Chicopee, MA, USA) equipped with an ESI source. The UHPLC-Q-TOF MS method used was performed as follows. The mass spectrometer was in positive ion mode. The ESI source conditions were set as follows: dry gas temperature, 320 °C; Sheath Gas Flow, 11 mL/min; Capillary Voltage, 3500 V; Collision Energies, 20 V, 40 V, 60 V; Sheath Gas temperature 350 °C; drying gas flow rate 8 L/min; nebulizing gas pressure 35 psi; mass range of 100–1100 *m*/*z*.

### 3.6. Identification of the Fractions’ Components

#### 3.6.1. Molecular Network

The Agilent Q-TOF MS/MS raw data in positive ionization mode were converted to the mzX ML format with the MS convert in ProteoWizard 3.0. Binary encoding precis at 32 bit and peak picking from MS levels at 1–2. The files were uploaded to GNPS (https://gnps.ucsd.edu/ProteoSAFe/static/gnps-splash.jsp (accessed on 6 April 2022)) with WinSCP. The files were used to create the MN with fragment ion mass tolerance of 0.5. The data were filtered by removing all MS/MS fragment ions within +/− 17 Da of the precursor ions. The MS/MS spectra were window-filtered by choosing only the top 6 abundant fragment ions in the +/− 50 Da window throughout the spectrum. The precursor ion mass tolerance was set to 2.0 Da, and the MS/MS fragment ion tolerance to 0.5 Da. A network was then created where the edges were filtered to have a cosine score above 0.7 and more than 6 matched peaks. The output files were visualized by MN with Cytoscape 3.8.2. GNPS online can provide a list of compounds and analyzed compounds by matching at least three fragments in the spectral library.

#### 3.6.2. Other Spectral Libraries and Literature

The MS/MS fragments were compared to the MS/MS fragments in a spectral library (PubChem: https://pubchem.ncbi.nlm.nih.gov/ (accessed on 15 April 2022)). The MS/MS fragments and possible fragmentation pathways were analyzed to identify the compounds of interest.

### 3.7. Statistical Analysis

Statistical analysis was conducted by Graph Prism 8.0.2 software, and the data from the experiment were analyzed by one-way ANOVA followed by Tukey’s post hoc multiple comparison tests between groups to compare anticonvulsant activities and showed as the mean ± standard error of the mean (S.E.M); *p* < 0.05 was considered statistically significant. The percentage of protection and convulsion was analyzed using a chi-square fisher exact test.

## 4. Conclusions

This study compared the anticonvulsant efficacy of different polarity fractions of BB extracts. The chloroform extract fractions exerted anticonvulsant activity with high potency in an animal model of the PTZ-induced convulsion. The anticonvulsant effect was likely due to the presence of different active compounds in the extract that exerted a synergistic effect. Sixteen compounds were analyzed, and the masses of these compounds were visualized by an MN approach. Lumichrome, pheophorbide A, and episyringaresinol 4′-*O*-beta-d-glucopyranose from BB were found for the first time. By comparing the chemical composition of the different polar fractions, we suggest, that aurantiamide acetate and loliolide that were only found in the chloroform extract fraction, are likely anticonvulsant active compounds. The molecular network approach provided a quick way to find novel compounds and represents a new approach for the study of effective compounds from TCM sources in the future. The results of this work provide data for further studies on the active compounds about anticonvulsant activity and a scientific basis for the treatment of epilepsy.

This research did not receive or use funds from any funding agencies in the public, commercial, or not-for-profit sectors.

## Figures and Tables

**Figure 1 molecules-27-08315-f001:**
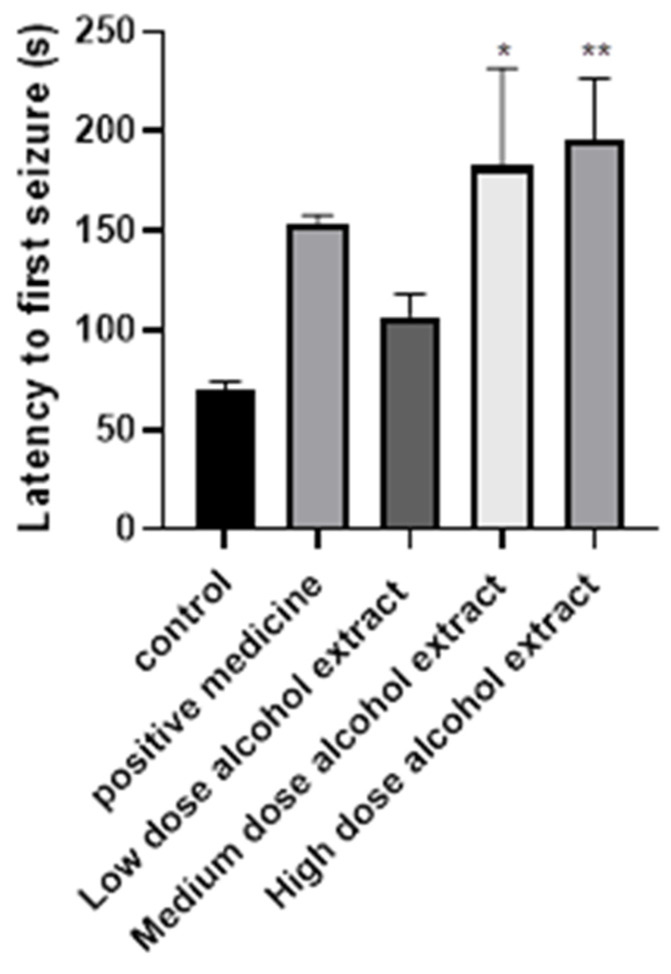
Latency time to first seizure. Data are expressed as mean ± SEM (*n* = 13/group). Statistical analysis was carried out by one-way ANOVA followed by Tukey’s test. * *p* < 0.05; ** *p* < 0.01.

**Figure 2 molecules-27-08315-f002:**
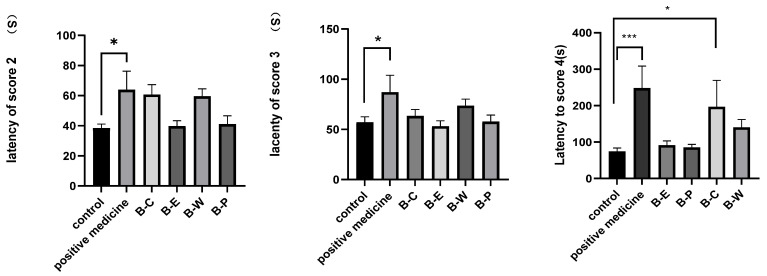
Latency time for score 2 induced by 80 mg/kg Pentylenetetrazollatency time for score 3; latency time for score 4. Data show the anticonvulsant effect of the samples after pretreatment with Pentylenetetrazol (80 mg/kg). Data are expressed as mean ± SEM (*n* = 13/group). Statistical analysis was carried out by one-way ANOVA followed by Tukey’s test. * *p* < 0.05; *** *p* < 0.001.

**Figure 3 molecules-27-08315-f003:**
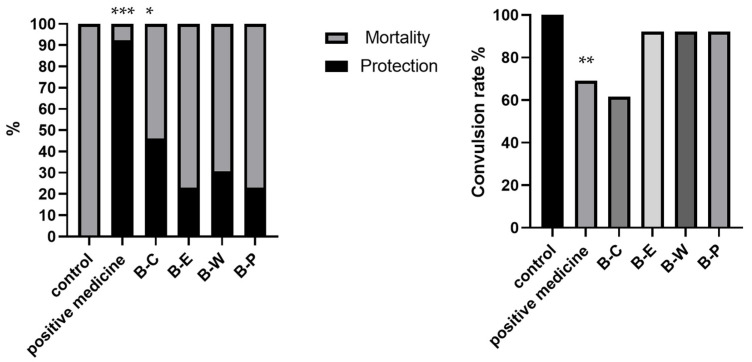
Chloroform extract fractions increased the percentage of protection and decreased the percentage of mortality and convulsion. Significantly different compared to the control group for percentage of mortality, protection, and convulsion using a chi-square Fisher extract test. * *p* < 0.05, ** *p* < 0.01, *** *p* < 0.001 significantly different from the respective vehicle-treated groups (control for positive effect and control for chloroform fractions).

**Figure 4 molecules-27-08315-f004:**
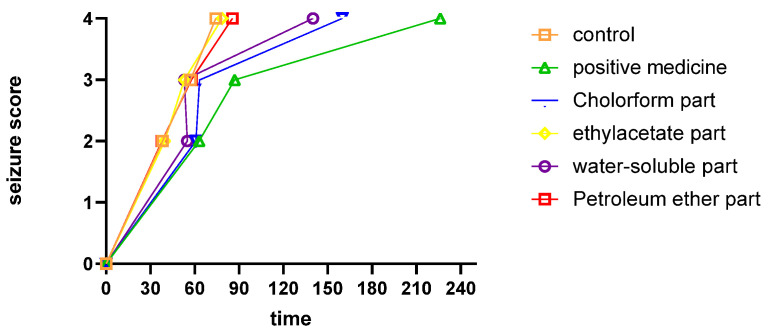
Relationship between convulsion score and latencytime based on seizure scores 1, 2, 3, 4.

**Figure 5 molecules-27-08315-f005:**
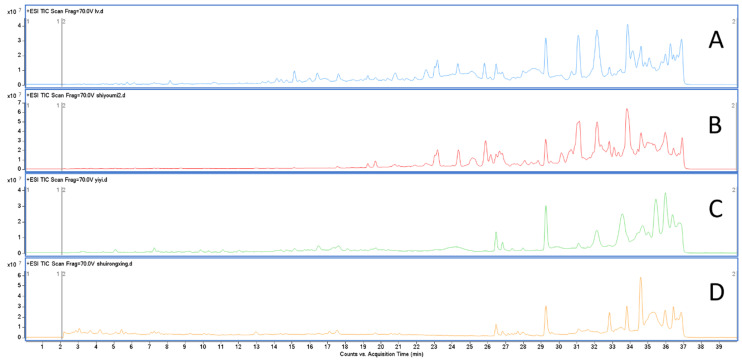
UHPLC-Q-TOF-MS/MS different polarity extract fractions of BB. Chloroform extract fractions (**A**), petroleum ether extract fractions (**B**), ethyl acetate extract fractions (**C**), water extract fractions (**D**).

**Figure 6 molecules-27-08315-f006:**
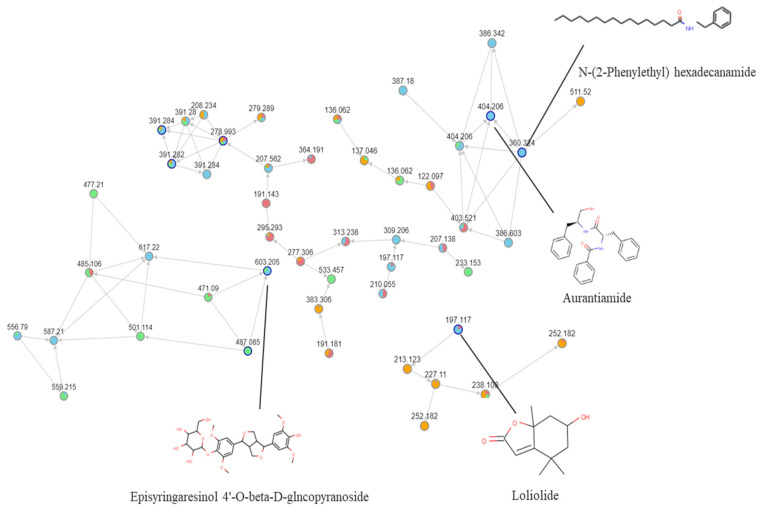
Molecular network based on metabolites from the different polarity fractions of the BB extract.

**Figure 7 molecules-27-08315-f007:**
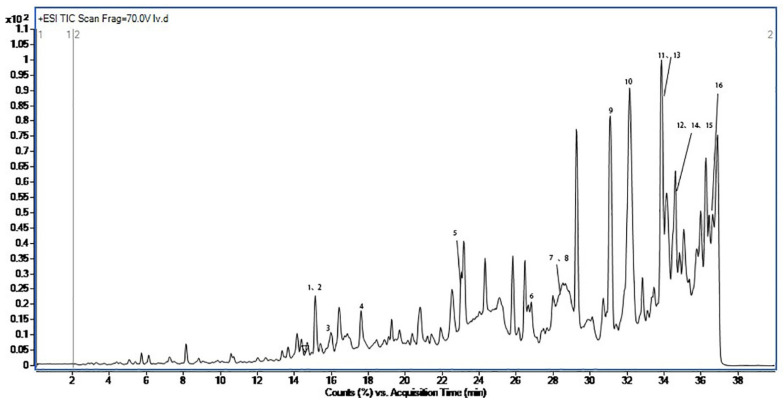
UHPLC-Q-TOF-MS/MS chloroform extracts of Bombyx batryticatus.

**Figure 8 molecules-27-08315-f008:**
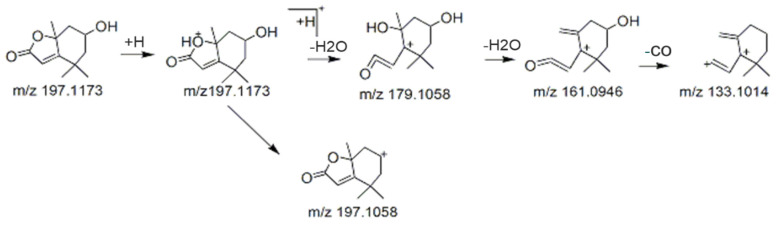
Fragmentation pattern of Loliolide.

**Table 1 molecules-27-08315-t001:** Effects of different extract fractions on latency time, percentage of protection, and percentage of convulsion for PTZ-induced seizures. * *p* < 0.05; ** *p* < 0.01; *** *p* < 0.001.

Treatments	Seizure Latency of Score 2 (s)	Seizure Latency of Score 3 (s)	Seizure Latency of Score 4 (s)	Protection	Convulsion
control	38.33 ± 9.65	57.09 ± 18.33	60.09 ± 31.19	0/13 (0%)	13/13 (100%)
Carbamazepine (80 mg/kg)	63.00 ± 18.53	94.17 ± 44.65	226.13 ± 34.87 ***	12/13 (92.31%) ***	9/13 (69.23%) **
B-C	60.71 ± 17.38	63.29 ± 17.41	159.88 ± 19.67 *	6/13 (46.15%) *	5/13 (61.54%)
B-E	39.70 ± 11.60	53.00 ± 18.49	78.50 ± 21.87	3/13 (23.08%)	12/13 (92.31%)
B-W	54.90 ± 20.33	73.21 ± 21.30	140.00 ± 39.34	4/13 (30.77%)	12/13 (92.31%)
B-P	41.00 ± 17.83	57.70 ± 21.02	95.20 ± 30.93	3/13 (23.08%)	12/13 (92.31%)

**Table 2 molecules-27-08315-t002:** Characterization of compounds in *Bombyx batryticatus* by UHPLC-Q-TOF-MS/MS.

Peak No.	tR (Min)	*m*/*z*	Fragment Ions	Adduct	Error (ppm)	Formula	Compound	Reference
1	15.1	197.1173	133.1014, 179.1058, 161.0946	M + H^+^	0.40	C11H16O3	Loliolide	[21,22]
2	15.2	331.1538	137.0594, 107.1292, 133.0642, 122.0346	M + H^+^	0.15	C19H22O5	3-(4-Hydroxy-3-Methoxyphenyl)Propyl 3-(4-Hydroxyphenyl)Propanoate	[23]
3	15.9	603.2046	185.0466, 441.1469, 425.1227	M + Na^+^	0.03	C28H36O13	Episyringaresinol 4′-*O*-beta-d-glucopyranose	___
4	17.8	243.0874	172.0859, 198.06941, 103.0547, 170.0714	M + H^+^	−1.04	C12H10N4O2	LUMICHROME	[24]
5	23.1	445.2135	194.1168, 224.1070, 105.0330, 252.1019	M + H^+^	2.96	C27H28N2O4	Aurantiamide acetate	[25]
6	26.5	279.1593	149.0238, 150.0247, 279.1593	M + H^+^	0.77	C16H22O4	Dibutyl phthalate	[26,27,28]
7	28.4	318.3004	282.2779, 270.2780, 264.2667	M + H^+^	0.14	C18H39NO3	Phytosphingosine	[29,30]
8	28.5	360.3240	105.0703, 122.0956, 360.3263, 106.0718	M + H^+^	−0.58	C24H41NO	*N*-(2-Phenylethyl) hexadecanamide	___
9	31.1	806.4002	134.1311, 244.1331, 784.4168, 262.1437	M + Na+	1.86	C45H57N3O9	Beauvericin	[31,32,33]
10	32.1	815.4594	244.1330, 134.0967, 537.2946, 262.1431	M + NH_4_^+^	−0.13	C46H59N3O9	Beauvericin A	___
11	33.9	926.6439	210.1490, 228.1595, 445.3109, 909.6165	M + NH_4_^+^	1.03	C48H84N4O12	Bassianolide	[34]
12	34.6	563.5511	282.2783, 265.2540, 247.2422, 283.2822	2M + H^+^	0.17	C20H39NO2	9-Octadecenamide, (Z)	[35,36]
13	33.8	256.2636	130.1232, 144.1374, 102.0909, 158.1541	M + H^+^	0.42	C16H33NO	Palmitamide	[37]
14	34.4	326.3060	308.2951, 309.2820, 121.1021, 135.1178	M + H^+^	1.97	C20H39NO	Oleoyl ethanolamide	[38]
15	34.8	593.2761	533.2545, 461.2311, 505.2225, 594.2779	M + H^+^	0.43	C35H36N4O5	Pheophorbide A	[39]
16	36.4	284.2954	284.2951, 285.2980, 286.3013	M + H^+^	2.14	C18H37NO	Octadecanamide	[40]

**Table 3 molecules-27-08315-t003:** Significance of the seizure score.

Score	Behavior
0	No response
1	ear and facial twitching
2	clonic jerk with hind limb extension
3	turning over onto side position, tonic-clonic seizures
4	clonic seizure with loss of righting reflex, generalized tonic–clonic seizures.

## Data Availability

The datasets used and/or analysed during the current study are available from the corresponding author on reasonable request.

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
