# Peer review of "Anticonvulsant Activity of Bombyx batryticatus and Analysis of Bioactive Extracts Based on UHPLC-Q-TOF MS/MS and Molecular Networking"

_molecules, 2022, doi:10.3390/molecules27238315_

Round 1

Reviewer 1 Report

The title

We can't claim identified based on the MS data. Please replace by annotated.

Introduction

Line 81: "The MN approach made us intuitively find relationships among compounds." what do you mean relationship? Can you please explain. 

line 83: "Then, we identified the active components in the fractions by UHPLC-Q-83 TOF MS and MN."We cannot apply the word identified to compounds by ms as we can not be sure of the identity. I advice to replace this word by annotated

Materials and methods

Please indicate the model of HPLC and of qTOF. 

In the raw mgf spectra the used detector is indicated to be ion trap, however you indicate in your manuscript to use qTOF. Can you clear this point please? 

Results

line 245 replace identified by annotated

In the chapter 3.3 please replace identified by annotated. On line 256 you report "comparison of retention times, MS spectra, and MS/MS spectral data with that in MN, Pubchem, other spectral libraries, and reference materials reported in the literature". How can you compare the retention times if you have not injected any of reference compounds in your hplc conditions?

from line 263 and lower for the retention time you can leave 1 digit after point. 

In the table you present some compounds that found to match with GNPS library. However, I suppose that these metabolites either contaminants or come from other sources. For exemple, Dibutyl phthalate is known contaminant coming from plastic. Pheophorbide A is the degradation product of chlorophyll, however I am not sure that this compound comes from Bombyx batryticatus, maybe from plants, used for their alimentation.

line 302. Please explain how can the retention time match with GNPS. Did you use the same chromatographic separation method, used in the library for each of library standards?

line 302. We can't claim identified, please replace by annotated

Please, change the text font of all latin names to italic.

Author Response

We would like to thank you for your careful reading, helpful comments and constructive suggestions, which have significantly improved the presentation of our manuscript. We appreciate the time and effort that you and the reviewers dedicated to providing feedback on our manuscript and are grateful for the insightful comments on and valuable improvements to our paper.  We have incorporated most of the suggestions made by the reviewers. Those changes are highlighted in the manuscript.  Please see a point-by-point response to the reviewers’ comments and concerns.

Reviewer 2 Report

In this work, the anticonvulsant effect of different fractions prepared from an ethanolic extract of Bombyx batryticatus (BB) is investigated. Among these fractions, the one obtained with chloroform showed a significant effect and its composition was investigated by LC-MS/MS and MN. Although the pharmacological results obtained in the PTZ-induced seizure model are clear and interesting, the molecular interpretation must be deepened to improve the article. For example, author should discuss some effects on the nervous system that has been reported for loliolide and aurantiamide acetate. It is noteworthy that the authors have not proposed a broader pharmacological network, including interaction networks of genes and proteins relevant to the pathophysiology of epilepsy, as illustrated in the following approach: https://doi.org/10.3389/fchem.2020.572952. The article provides relevant information, since insects are attracting interest as an economic source of new bioactive molecules.

Specific Comments

·         Kindly check the wording of the penultimate paragraph of the introduction: Many compounds of BB were isolated, then identified….

·         The following sentence is not clear: ….1mL solution used for pharmacological experiments was dissolved in 15 mL methanol and filtered through a 0.22 -μm microporous membrane filter after centrifugation. Is this dilution of the extracts in methanol supposed to be the one used for LC-MS/MS analysis?

·         Based on what criteria the doses of the extracts were selected from 10-30 g/kg before administering them to the mice.

·         Please clarify this sentence: The mobile phase was a binary solvent system consisting of solvent A (5 mM ammonium…acetate, formiate, or is 5 mM ammonium hydroxide?

·         The description of the MS/MS fragmentation analysis (Figure 7) of the chloroform extract requires rewriting. It is suggested to follow the order of table 3. Some retention times simply do not match leading to confusion. Why was MS analysis not performed in negative mode ion?

·         …..After PTZ administration, all animals in different groups were placed in chambers (please, specify the size)  and their behavior was recorded for 30 minutes.

·         Various plant names are abbreviated or de-italicized. For example….L271-272:……… N.glauca and D. stramonium as alternative hosts to N. tabacum, P. peruviana and S. origanifolia.

·         Explain why does diethyl phthalate appear in the chloroform extract?

·          Some graphs do not have units of time. For example, the seconds (s) do not appear in the first two graphs of Figure 2

·         Figure 5. The description is not clear and does not allow the identification of the fractions of the chloroform extract. The color coding also does not match the description in the text from the results and discussion section.

Author Response

(The authors gave the same response as above.)

Round 2

Reviewer 1 Report

Dear Authors, 

Thank you for your modifications. However, still the identification term persist in many cases. Can you please modify?

Line 23, 29, 253, 266, 302, replace  "identified" by "annotated". Without the retention time we can't claim that we have identified the compound.

Reviewer 2 Report

After carefully reading the new version of the article, I find the answers and corrections quite satisfactory. The work improved significantly and it can be published.